# The Role of Dairy in Human Nutrition: Myths and Realities

**DOI:** 10.3390/nu17040646

**Published:** 2025-02-11

**Authors:** Laura Sanjulián, Salvador Fernández-Rico, Nerea González-Rodríguez, Alberto Cepeda, José Manuel Miranda, Cristina Fente, Alexandre Lamas, Patricia Regal

**Affiliations:** Food Hygiene, Inspection and Control Laboratory (Lhica), Department of Analytical Chemistry, Nutrition and Bromatology, School of Veterinary Science, Campus Terra, University of Santiago de Compostela, 27002 Lugo, Spain; laura.sanjulian@usc.es (L.S.); salvador.fernandez.rico@usc.es (S.F.-R.); nerea.gonzalez@usc.es (N.G.-R.); alberto.cepeda@usc.es (A.C.); cristina.fente@usc.es (C.F.); alexandre.lamas@usc.es (A.L.); patricia.regal@usc.es (P.R.)

**Keywords:** cow milk, dairy, diet, human health, probiotics, fermented milk

## Abstract

Milk is a highly complex food that contains all the nutrients necessary for the development of mammalian offspring. For millennia, humans have included milk and milk products as major components of their diet. However, the effect of the consumption of dairy products on health has been a concern in recent years in terms of myths and realities. This review briefly describes the composition of bovine milk, the positive and negative effects that have been related to dairy products, and those aspects where the scientific evidence is still inconclusive. In addition to being nutritional, dairy products are a source of bioactive peptides, prebiotics and probiotics, fatty acids such as CLA, and fat globule membranes or have a protective effect against certain diseases. Negative effects include milk protein allergy or lactose intolerance. The effects of dairy products on certain cancers, such as prostate cancer, and their role in type II diabetes mellitus or weight gain are still inconclusive. Although the role of dairy products in cardiovascular risk is still inconclusive, recent meta-analyses have shown that dairy products may have a protective effect.

## 1. Introduction

In the Codex Alimentarius [1], milk is defined as “the normal mammary secretion of dairy animals obtained by one or more milking’s without any type of addition or extraction, destined for consumption in the form of liquid milk or further processing” [1]. Milk is one of the oldest and most traditional foods consumed by men. The habit of milk and dairy product consumption in human nutrition dates back to the origins of evolution, with milk and dairy products being among the staple components of the normal diet today and contributing 15% of the total food intake in European countries [2].

Despite its evident importance, over the last 20 years, the consumption of cow milk has declined in Western European countries and the US [3]. This is because consumers have begun to question milk as a perfect food for health, taste, or animal welfare and environmental protection reasons [3]. Curiously, the consumption of dairy products such as cheese remains stable or increases, which may be due, among other things, to the fact that there are no plant-based alternatives, such as milk [3]. Among health problems, many people associate digestive problems with milk consumption and lactose intolerance (LI). However, most people who stop consuming this food for this reason do so without consulting a doctor or undergoing a lactose intolerance test [4]. The symptoms of cow milk protein allergy (CMPA) and lactose intolerance are considered the main adverse effects of milk and milk product consumption [5]. The terms “milk allergy”, “milk intolerance”, and “lactose intolerance” are frequently employed without a precise comprehension of their distinct meanings, the diverse mechanisms underpinning each condition, or the dietary implications associated with their diagnosis. LI treatment generally involves a marked reduction but not complete elimination of lactose-containing foods. In contrast, individuals suspected of having CMPA should undergo strict cow milk avoidance [6]. These misleading statements have cast doubts on the nutritional integrity of milk, fostering confusion among consumers and challenging the reputation of these essential foods. To elucidate the actual impact of milk on human health, a reliance on accurate, rigorous, and evidence-based information is imperative. This review aims to explore key facets of the nutritional composition of milk, offering insights into its real role in promoting human health. By scrutinizing studies with a high level of scientific evidence, this review seeks to dispel myths and contribute to a clearer understanding of the intricate relationship between milk consumption and human well-being.

## 2. Nutritional Composition of Milk and Dairy Products

Milk is a complex biological fluid, in part owing to the great variety of nutrients and biological components it provides but also due to its vital role in the development of the newborn. Its composition is summarized in Table 1.

Milk is primarily composed of water, fat, proteins, lactose, and salts—constituents that not only define its nutritional profile but also contribute significantly to its commercial value. In addition to these fundamental components, milk contains a myriad of other constituents present at relatively low concentrations, including vitamins, minerals, enzymes, bioactive peptides, cytokines, hormones, dissolved gases, and various lipids, such as conjugated linoleic acid (CLA) and microribonucleic acids (miRNAs). These additional elements play pivotal roles in shaping the biological and technological properties of milk, contributing to its diverse range of functionalities [7,8,9,10]. As milk is one of the most studied food products at all times, there is a significant amount of scientific data concerning its nutritional composition. However, other constituents, such as bioactive peptides, miRNAs, and enzymes, among others, have not been studied in such detail thus far.

### 2.1. Milk Proteins

Within the nitrogenous fraction, milk contains between 3% and 4% of these substances, which are distributed as caseins (78% of the milk nitrogen), whey proteins or serum proteins (17% of the milk nitrogen), and nonprotein nitrogen (5%) [7,8].

Caseins can be subdivided into five types, caseins αs1, αs2, β, γ and κ, which are grouped to form polymeric molecular complexes called casein micelles. Among whey proteins, the most abundant is β-lactoglobulin, which accounts for 50% of them; α-lactalbumin, serum albumin, protein-peptones, immunoglobulins, metalloproteins, such as lactoferrin, transferrin or ceruloplasmin; and enzymes, i.e., lipases, proteases, or phosphatases, also belong to this group [7].

Dairy proteins have high biological value and contain all the essential amino acids, highlighting the high content of branched amino acids (isoleucine, leucine, and valine), particularly lysine. Table 2 presents a comparison between the amino acid profile of the protein pattern for children aged 1–3 years and milk protein [7]. It has been reported that milk covers the amino acid needs for child development. These proteins not only are proteins of high biological value but also have a digestibility of approximately 95%, and their net protein utilization rate (NPU) is similar to that of egg protein, ranging from 85–90% [11].

### 2.2. Milk Fat

Milk fat is a complex nutrient constituted fundamentally by saponifiable lipids (99%), plus a minority of unsaponifiable fractions (1%). These lipids form emulsified spherical micro globules in the milk aqueous phase called fat globules, and this disposition favors the digestive enzymes acting on them and their subsequent hydrolysis [10,12].

More than 400 different fatty acids have been identified in milk, although most are present in amounts less than 1% [13]. The fatty acid composition of bovine milk fat depends significantly on the feeding and milk yield of the animal, the genetic constitution, and the race of the individual, as well as the lactation stage [14,15,16,17,18]. Those quantitatively most important are short- and medium-chain fatty acids (up to 10 carbon atoms), with an even number of carbons, unbranched and saturated. Saturated fatty acids represent 68.8% of the total fatty acid content (by weight), with myristic, palmitic, and stearic fatty acids being the most abundant. Monounsaturated acids represent approximately 27.3% of milk fat, with oleic acid being present in significant quantities. Polyunsaturated compounds constitute 3.97%, especially linoleic acid (LA) and α–linoleic acid (ALA), as do EPA and DHA.

In bovine milk and bovine meat, small amounts of trans MUFAs (3.1%), trans PUFAs (0.034%), and cis/trans and trans/cis PUFAs (1.34%) have different biological effects [19]. One of these, trans-vaccenic acid (trans-11 C18:1, or trans-C18:1 *n*-7, TVA), directly promotes effector CD8+ T-cell function and antitumor immunity in vivo [20]. In an in vitro study, Song et al. [21] reported that TVA inhibits proliferation and induces apoptosis in human nasopharyngeal carcinoma cells. Circulating TVA in humans comes primarily from ruminant-derived foods, including dairy products such as milk and butter [22]. Although the levels of TVA in cow milk are usually lower than those in human milk, modifying the diet of cows can increase the levels of this fatty acid [23]. Some short-chain fatty acids, such as butyric acid (C4:0), or odd-chain fatty acids, such as pentadecanoic (C15:0) and heptadecanoic (C17:0), are present only in the fat of ruminants. Therefore, its presence in subcutaneous adipose tissue or in human serum constitutes an indicator or biomarker of milk fat intake [13]. Table 3 presents a summary of the major fatty acid distributions in bovine milk. Another component of milk fat is polar lipids (PLs). They are located mainly in fat globules. The main PLs in milk are phosphatidylcholine (PC), sphingomyelin (SM), and diacyl- and plasmalogenic phosphatidylethanolamine (PE), which occur in similar proportions. To a lesser extent, cow milk also contains phosphatidylinositol (PI), phosphatidyl serine (PS), and glycosphingolipids. PLs represent 1% of milk fat, which corresponds to approximately 29.4–40 mg/100 g of raw cow milk [24,25,26]. Despite being present in low proportions, LPs have been shown to have a positive effect on neurological development, inflammation, cardiovascular disease, cholesterol absorption, and stress [27].

In the unsaponifiable fraction, cholesterol, β-carotene and fat-soluble vitamins (A, D, E and K) are present. Cholesterol is the main sterol of milk, representing at least 95% of the total sterol content, ranging from 271.37 mg/100 g of fat in conventional milk to 278.76 mg/100 g of fat in organic milk [28]. Other sterols, such as β-sitosterol, campesterol, and stigmasterol, may also be present in milk derived from the diet of the animal [29].

### 2.3. Milk Carbohydrates

Lactose is an exclusive disaccharide of milk that, in bovines, is the major milk solid and milk solid-n nonfat, representing approximately 4.70% of milk [30]. Scientific knowledge about the physiology and variability of lactose is still scarce because of the consensus that this compound is constant in milk and does not affect milk quality or technological properties [31]. In addition to lactose, small amounts, less than 100 mg/L, of other sugars (glucose and galactose), glycoproteins, and glycolipids are found in the glucidic fraction of milk. The oligosaccharides present in cow milk deserve special mention. This product contains between 30–60 mature oligosaccharides [32]. The concentration of cow milk oligosaccharides varies according to the time of lactation. In colostrum, the values are approximately 1–2 g/L, whereas in mature milk, they are found at a concentration of 30–60 mg/L. The production of these molecules decreases with the lactation period until it stabilizes in late lactation. Among the ten oligosaccharides present in cow milk, 50% are 6′-sialyllactose (6′-SL) and 3′-sialyllactose (3′-SL) [33,34]. In addition to lactation time, other factors, such as breed, influence their concentration. Holstein milk has higher concentrations of HexNAc-lactose and 6′-SL, whereas Jersey milk has higher quantities of fucosylated and sialylated oligosaccharides [35]. They have been identified as compounds with valuable physiological and metabolic effects in vivo. In addition to a prebiotic effect, evidence since the beginning of the century has suggested that oligosaccharides from milk are partially absorbed in the intestines and contribute to the development of molecular structures in the brain [33,36,37,38], have anti-inflammatory effects, reduce the adhesion of pathogens, prevent respiration, and contribute to the development of the mucosa and immune system [32].

### 2.4. Mineral Milk Fraction

The mineral salt content represents approximately 1% of milk, both dissolved and in the colloidal state, forming associations with caseins. Milk and dairy products are generally excellent sources of minerals, especially calcium, which can reach concentrations of 1.244 mg/100 mL of milk, as shown in Table 1. For example, on the basis of dietary reference values (DRVs) of the EFSA [39], adult women require 750 mg/day. Therefore, a cup of milk (250 mL) would provide 40% of the daily calcium requirement. However, phosphorus, zinc, sodium, potassium, iodine, selenium, and chromium are also relevant constituents of dairy. The phosphorus content was 92 mg/L according to Table 1. The adequate intake recommended by the EFSA DRV is 550 mg/day. Therefore, a cup of milk provides half of the daily adequate intake. For example, it is a good selenium source and a remarkable contributor to intake [40]. However, milk is an iron-deficient food [7].

### 2.5. Vitamin Content in Milk

The vitamin fraction of milk is composed of lipophilic (A, D, E, and K) and hydrophilic (B1, B2, B3, B5, B6, B8, B9, B12, and C) vitamins. Whole milk is considered a good source of vitamins A and D, with concentrations of 56 µg/100 g and 0.05 mg/100 g, respectively. The amount of milk fat and other variables, such as the animal’s diet and the season of the year, influence it [41].

The daily intake of dairy products in the form of whole milk contributes significantly to the satisfaction of the vitamin requirements of an adult and can reach 22 to 32% of the need for vitamin A, 34 to 100% of the need for vitamin D, 64 to 92% of the need for vitamin B2, 34 to 42% of the need for vitamin B5, and 84 to 112% of the need for vitamin B12 [41,42].

## 3. Positive Effects of Dairy Consumption

### 3.1. Calcium Source

The enormous importance of calcium for humans lies in the wide number of physiological functions in which it affects the growth and development of the skeleton and teeth, bone mineralization, nerve conductivity, muscle contraction, hormonal secretion, and enzyme or blood clotting. When calcium consumption is analyzed in detail, poor consumption is observed in all population groups, especially in elderly individuals, and specifically in women. This deficient consumption is associated with age-related bone loss and increased risks of osteoporosis, fractures, hypertension, cardiovascular disease (CVD), and colon cancer [43].

There are several food groups that provide adequate amounts of calcium to humans, such as nuts, green leafy vegetables, legumes, and milk or dairy products. However, the latter are considered the best dietary source of calcium, both due to the concentration in which it is found, reaching between 55% and 70% of the daily requirements (Table 4), as well as the absence of factors that inhibit calcium absorption, such as its high bioavailability, this term referring to the dietary calcium fraction that is potentially absorbable by the intestine and whose purpose is to be employed in physiological functions. Although it is theoretically possible to cover calcium recommendations with a dairy-free diet, most studies agree that their total elimination makes it difficult to adequately cover the recommended intake of this mineral in all stages of life [44,45]. Once it reaches the intestine, the calcium of the diet is mixed with digestive secretions and is absorbed mainly in the upper part of the ileum by two mechanisms: passive paracellular diffusion and active transcellular transport. The latter requires energy and is mediated by 1,25-dihydroxyvitamin D or calcitriol.

Passive paracellular diffusion is a nonsaturated process that is independent of vitamin D and age and increases with increasing dietary intake if calcium is in an ionic form (Ca^2+^) or forms complexes with organic molecules. Certain components of the diet favor the solubility of calcium, including phosphopeptides from casein and amino acids such as lysine or arginine, which form chelates with calcium and stimulate passive diffusion, and some carbohydrates, especially lactose, which significantly increase the passive absorption of calcium in the absence of vitamin D. Milk and dairy products therefore meet the ideal conditions for the use of calcium by humans. Casein contributes to the intestinal absorption of calcium, which is produced during its digestion of soluble and easily absorbable compounds called casein phosphopeptides (CPPs). Phosphorylated peptides from α1, α2 caseins, and ß-casein can form soluble complexes with minerals such as calcium but also with iron and zinc at the intestinal pH, influencing their bioavailability. These mineral-carrying peptides, CPPs, can be released during the enzymatic digestion or processing of dairy products. These formations are able to bind to calcium, resulting in soluble complexes that prevent the precipitation of calcium phosphate in the intestine and increase its absorption and bioavailability [44,45]. Although the mechanism is not entirely clear, lactose increases the transport of calcium through the nonsaturated paracellular pathway, specifically by increasing the volume of distal intestinal fluid, increasing the intercellular space and, consequently, the permeability of the intestinal mucosa. However, the amount of lactose that milk provides does not seem to significantly influence the absorption of calcium in healthy adults, and even if it does, it may be hidden by active transport, which, under conditions of moderate calcium intake and adequate levels of vitamin D, is sufficient.

Passive transport can be relevant in situations where active transport is compromised, such as vitamin D deficiency, or when high intake of calcium is needed, as is the case for infants and elderly individuals, where solubility is a limiting factor and passive absorption the main mechanism [44]. The Ca/P ratio of milk is also beneficial for absorption and is ideal between 1 and 1.50. A value greater than 1.50 in the diet is associated with an increase in renal calcium excretion [44].

As mentioned above, numerous foods stand out because of their high calcium content; however, few of them constitute a good source of the element in question, either because they are consumed in very small quantities, as is the case for some aromatic herbs (basil, thyme, and dill) or spices (cinnamon), or because of the presence of dietary components (unabsorbed lipids) or intrinsic food compounds such as cereal and seed phytates, spinach oxalates, walnuts and sorrels, tea tannins, and partially methoxylated pectin of vegetables. The latter compounds that are not present in dairy products inhibit the intestinal absorption of calcium by forming poorly soluble combinations between them. Thus, we find that despite having calcium contents equal to or higher than those of dairy products and with usual serving sizes, food presents lower absorption rates [44,46]. As shown in Table 4 it is very difficult for the Western population to obtain enough calcium following a dairy-free diet unless they incorporate calcium-enriched food and/or supplements into their diet.

### 3.2. Source of Bioactive Peptides

The first record of bioactive milk peptides dates back to 1950, when Mellander [47] reported that the ingestion of phosphorylated peptides derived from caseins increased calcification via a route independent of vitamin D in children with rickets [8]. Molecules such as lysozyme, lactoferrin, immunoglobulins, or growth factors, which are naturally present, are encrypted and inactivated in the primary structure of milk proteins and are released after enzymatic hydrolysis or microbial fermentation of casein (α, β, γ, and κ casein) and whey proteins (β-lactoglobulin, α-lactalbumin, serum albumin, immunoglobulins, lactoferrin, and protein-peptone). They can also be obtained during processing or ripening [8,48,49].

For a bioactive peptide to exert its activity following oral administration, its active form must resist digestion, be absorbed through the gastrointestinal (GI) epithelium via transcellular or paracellular pathways, and reach the target organs either intact or as bioactive metabolites. These bioactive peptides are related to numerous health effects (Figure 1), ranging from antithrombotic, antihypertensive, and inflammatory to antioxidant, antimicrobial, or antiobesogenic properties. Some of them are multifunctional and have two or more qualities, which means that they can act at virtually any level: cardiovascular, digestive, endocrine, immune, or nervous. Their action on one system or another system depends on their composition and amino acid sequence [48,50,51].

Notably, certain peptides, fragments of caseins and whey proteins called casoquinines and lactoquinines, respectively, which behave as inhibitors of angiotensin-converting enzyme (ACE), have been identified in milk and cheese [48]. These peptides, which are called antihypertensive peptides, act as competitive inhibitors of ACE, avoiding the synthesis of angiotensin II, a potent vasoconstrictor, and lowering blood pressure [8]. In moderately hypertensive individuals, daily intake of 30 g of Grana Padano for two months reduces blood pressure [52]. On the other hand, milk-coagulating proteins, such as chymosin and κ–casein, have mechanisms similar to those of blood coagulation, which are related to thrombin and fibrinogen, and are able to suppress platelet aggregation by occupying the binding site to the platelet receptor and inhibiting fibrinogen, interfering with the synthesis of thrombin; in other words, they have antithrombotic properties [8,49].

Milk proteins and peptides are also potent antioxidants that inhibit reactive oxygen species (ROS), act as prooxidant metal scavengers, and decrease hydroperoxide levels. The protein fractions of whey are rich in tyrosine (Tyr), tryptophan (Trp), methionine (Met), lysine (Lys), cysteine (Cys), and histidine, which are examples of amino acids related to antioxidant activity [53].

The antimicrobial peptides present in milk can inhibit a wide range of pathogenic bacteria, such as *Listeria*, *Salmonella*, *Escherichia*, *Staphylococcus*, or *Helicobacter*, filamentous fungi, and yeast. The antimicrobial potential of milk is linked to immunoglobulins, other proteins, such as lysozyme, lactoferrin, and the lactoperoxidase system, and protein fractions, which are the precursors of those peptides that stimulate the innate immune system of the organism against the attack of pathogens. Milk caseins are an important source of antibacterial peptides [51].

Milk proteins and protein fractions also have a certain anticancer potential. The case of peptides derived from whey proteins is due to the presence of γ-glutamylcysteine and cysteine or cysteine dipeptides, which function as precursors in the biological synthesis of glutathione. β-Lactoglobulin and α-lactalbumin decrease the toxic effects of cancer. Smaller fractions, such as lactoferrin, have shown significant results in the fight against intestinal tumors, promoting apoptosis, inhibiting angiogenesis, modulating the metabolizing enzymes of carcinogens, and acting as iron sequestrants. The peptides derived from the hydrolysis of caseins also have antimutagenic properties and potent inhibitory activity against mutagens; however, the mechanisms have not yet been elucidated [8,49].

Peptides from cheese enhance the human immune system by modulating immune cell responses and regulating inflammatory cytokine production. In particular, βP-casein-derived compounds stimulate macrophage phagocytosis and enhance mouse lymphocyte proliferation. Notably, fragments 192 to 209 of ß-casein-derived peptides have been shown to modulate immune functions. This immunomodulatory activity extends to various cheeses, including Parmigiano Reggiano, Chinese Rushan, Naizha, Goatskin Tulum, Edam, Gouda, Karish, and Ras [49].

### 3.3. Source of Prebiotics and Probiotics

The main components of dairy products with prebiotic functions are lactose and oligosaccharides. Lactose, which is not digested by lactase in the small intestine, arrives intact to the colon, where lactic acid bacteria internalize it through phosphotransferase systems in the form of lactose-6-phosphate, which, via the action of phospho-β-galactosidase, is transformed into glucose and galactose-6-phosphate. Glucose activated in the form of UDP-glucose and UDP-galactose is used in the production of EPS, which acts as an excellent substrate for the growth of bifidobacteria and lactobacilli, favoring the autochthonous intestinal microbiota and inhibiting colonization by enteropathogenic bacteria sensitive to bacteriocins, lactic acid (acid pH), and metabolites such as short-chain fatty acids (SCFAs). Some of the products of fermentation, such as CH_4_, H_2_, and CO_2,_ can trigger gastrointestinal symptoms, which are termed lactose intolerance and can depend on the quantity of lactose ingested [54].

Although bovine milk contains fewer oligosaccharide structures than human milk (HMOs), these structures are typically considered indigestible by human enzymes and can be considered potential prebiotics. Instead, lacto-N-tetraose and the sialyllactose isomer pair, prebiotic oligosaccharides from human milk, can also be found in cow milk [55]. They stimulate the growth of beneficial microorganisms, mainly of the *Bifidobacterium* genus (dominant species in breastfed infants), and, to a lesser extent, some strains of *Lactobacillus*. As these bacteria specifically express sialidases and fucosidases, it is believed that HMOs select these strains since other bacteria are not able to use them [56,57,58]. The prebiotic effect of bovine milk oligosaccharides, which contribute to the increase in bifidobacteria in the intestinal tract, has been identified as an important factor contributing to the prevention of obesity. The presence of bifidobacteria has also been correlated with a reduction in diabetes symptoms, including increased glucose tolerance [59]. Therefore, modulation of the gut microbiota may be another promising strategy to prevent these prevalent metabolic problems. The regulation of lipid and glucose metabolism by the consumption of milk oligosaccharides and *Bifidobacterium infantis* in genetically predisposed animal models has been suggested [60]. In a similar study carried out in mice with diet-induced obesity, bovine milk oligosaccharides in combination with *Bifidobacterium longum* decreased gut permeability, restored the intestinal microbiota, and improved inflammation. This combination increased the levels of *Lactobacillus* and restored the *Allobaculum* and *Ruminococcus* levels [61]. In a preclinical neonatal model, the addition of bovine milk oligosaccharides increased the abundance of *Bacteroides* in the ascending colon [62]. Industrial interest in commercializing these compounds for therapeutic purposes is increasing. However, the essential relationship between their structure and function has yet to be fully elucidated. Membrane filtration and various synthesis strategies for milk oligosaccharides (OSs) can enable their production for use as ingredients in therapeutic foods, thereby increasing the availability of milk-derived OSs and their bioactivities for broader public consumption [33].

The majority of microorganisms used in the development of probiotic products are lactic acid bacteria (LAB) and bifidobacteria. Dairy products, particularly yogurt and fermented milk, serve as ideal vehicles for delivering or producing active live cultures. The buffering effect of milk proteins helps preserving the viability of probiotics as they transit through the gastrointestinal (GI) tract, where they exert their beneficial effects. The impact of fermented milk containing LAB depends on various factors, including the specific strain of lactic acid bacteria used in fermentation, the presence of one or more types of LAB, potential interactions between strains, the type of fermented product, the timing of milk consumption, the individual’s genetic constitution, the presence or absence of disease, and the administered dose [63]. Fermented milk products have been shown to have a positive effect on the gut microbiota. These bacteria are a source of beneficial bacteria. In these products, the food matrix protects probiotics during their passage through the gut [64]. For example, kefir has a dual beneficial effect since not only is it a source of microorganisms but the bacterial exopolysaccharides formed during fermentation also favor an increase in the abundance of *Bifidobacterium* [65]. Similarly, the consumption of this fermented product increases the intestinal levels of *Lactobacillus* [66].

### 3.4. CLA Source

CLA is a type of trans fatty acid that, unlike most trans fatty acids, has been shown to possess physiological properties that are beneficial to health due because of its anticarcinogenic, antiobesogenic, antidiabetic, and antihypertensive activities. CLAs are family of at least 28 isomers of linoleic acid derivatives synthesized after the action of the enzyme isomerase-LA, which converts dietary LA into vaccenic acid (biohydrogenation) in the rumen, which subsequently becomes CLA after being attacked by the Δ9-desaturase enzyme of the mammary gland. The second route through which CLA can be produced is as an intermediate product of the microbial biohydrogenation of LA to vaccenic acid in the bovine rumen [8,67].

The presence of CLA is therefore typical of products derived from ruminant animals, such as meats (with 25% CLA intake) and dairy products (with 70% CLA intake) [67]. The CLA content of cow milk depends fundamentally on the animal’s feed and varies within a broad spectrum ranging from 2 to 37 mg/g of fat, with cis-9, trans-11 CLA, called ruminal acid, as the predominant isomer [68], which has been associated with anticarcinogenic [67], anti-inflammatory, antiatherosclerotic and antiobosogenic effects derived from its antioxidant properties [69]. Although there is no recommended intake of CLA, with an intake of 3–6 g/day, the beneficial effects of CLA are ensured [70].

A recent comprehensive review compiled information on the positive health outcomes associated with CLA. Research indicates that CLA can contribute to the improvement of various health issues, exerting effects on obesity, inflammation, anticancer properties, atherogenicity, immunomodulation, and osteosynthesis. It is considered a nutritional pathway for enhancing lifestyle and addressing metabolic syndrome. The primary contributors to these effects are the two main isomers of CLA: the cis-9,trans-11 isomer (c9,t11) and the trans-10,cis-12 isomer (t10,c12), along with the CLA mixture [71]. A meta-analysis that evaluated the effects of CLA supplementation on blood pressure and endothelial function in adults revealed that CLA supplementation did not significantly alter systolic or diastolic blood pressure but significantly reduced the level of intercellular adhesion molecule (ICAM), which is implicated in the attraction of leukocytes to the vascular endothelium [72]. Another meta-analysis revealed that CLA supplementation has a beneficial effect on the levels of adipokines and cytokines, including serum IL-6 and leptin, but increases fasting blood glucose and aspartate aminotransferase [73]. Importantly, the authors of this review indicate that the effects are small and may not reach clinical significance. Another meta-analysis focused on the combined effects of CLA and exercise on body composition and obesity and reported that this combination resulted in a reduction in body fat and insulin resistance but did not reduce body weight [74].

Conversely, adverse effects linked to CLA have also been documented, including impacts on glucose homeostasis, insulin resistance, hepatic steatosis, and the potential induction of colon carcinogenesis in humans. Most of these risks are attributable to the t10,c12-CLA isomer, whereas c9,t11-CLA may be without hazard for human consumption, which must be confirmed as well [75]. Moreover, several adverse effects have now arisen that are not yet completely understood because of the lack of human studies and the absence of enough scientific information to determine whether these adverse effects are related to the CLA dosage or duration of administration. Since consuming too many CLAs through supplements can have negative health consequences, it is best to obtain them naturally, and milk and fermented dairy products are good sources for taking advantage of their benefits [71]. The CLA content naturally present in milk is not very high, which makes it impossible to consider it as a health promoter because of its CLA content. However, it has been discovered that modifying a cow’s diet by introducing seeds, such as flax, can increase CLA levels in milk, and these products can then be considered relatively important sources of CLA. In no case do you reach the potentially toxic levels that the habitual consumption of supplements would imply.

### 3.5. Osteoporosis Prevention

The World Health Organization (WHO) (Organization 2003) defines osteoporosis as “a systemic disease, characterized by a decrease in bone mass and a deterioration of the microarchitecture of the same, with the consequent increase in the fracture risk”, which is a major public health problem, as it is associated with high morbidity and mortality. In humans, bone mass gradually increases until 20–30 years of age, when the maximum peak is reached, which compensates for the losses that the bone will suffer thereafter; therefore, the risk of osteoporosis decreases with increasing age [76]. Milk and dairy products are recognized globally as promoters of bone health, largely because they are sources of high-bioavailability calcium but also because of the contributions of other elements, such as minerals, peptides, or CLA, which act positively on bone mass, the prevalence of fractures, and the prevention of osteoporosis [41,77].

Numerous observational studies have shown the favorable effect of dairy products on bone mass during childhood and adolescence, which are fundamental life stages in the acquisition of bone tissue to ensure adequate peak bone mass (PBM). A greater beneficial effect of dairy intake than of calcium intake in the form of supplements is observed in some cases. A dairy-free diet has also been linked to lower PBM and a higher rate of fractures in children [78]. However, it is important to note that meta-analyses do not yield clear results concerning the relationship between dairy consumption and bone problems. For example, in Europeans and non-Hispanic Whites from North America, dairy consumption was not clearly associated with osteoporosis, hip fracture risk, or bone mineral density, but a diminished risk of vertebral fracture was observed [79].

The results regarding BMD changes were heterogeneous and did not allow for a definitive conclusion.

### 3.6. Preventive Factors in CRC and Other Cancers

Milk consumption is a modifiable lifestyle factor that has been associated with several types of cancer in observational studies. However, there is limited evidence on the causality of these relationships. Focusing on the protective role of milk consumption against colorectal cancer, a review performed in 2022 confirmed that vitamin D supplementation seems to have a protective effect on the prevention and treatment of colorectal cancer (CRC), whereas calcium intake has contradictory effects as a prevention or treatment tool [80]. A review of prospective studies and randomized clinical trials indicates that calcium may inhibit colorectal carcinogenesis by acting on proliferation, differentiation, and cell apoptosis and forming complexes with free fatty acids and bile acids. Moreover, calcium intake below 700–1000 mg/day increases the risk of CRC, so increasing calcium intake levels to within this range is recommended [81]. Vitamin D exerts anti-inflammatory effects through bile acid catabolism, suppresses the signals of the nuclear factor enhancer of the kappa light chains of activated B cells (NF–κB) and cyclooxygenase-2 (COX-2), and increases the production levels of interleukin-10 (IL-10). However, no recommended intake was established due to the inconsistency of the studies consulted [81].

The relationships between dietary intake of calcium/vitamin D and CRC risk have been addressed by several meta-analyses based on cohort studies and clinical trials. Higher dietary intakes of calcium and vitamin D are associated with a decreased risk of CRC [82]. Milk is a natural source of calcium and vitamin D, and these nutrients synergistically interact in the human body. Dairy products and milk have been associated with lower risks of CRC, perhaps related to their high calcium content [82,83,84]. The study by Song et al. also reviewed the role of dairy as a preventive factor due to its high content of calcium, micronutrients, and bioactive compounds, establishing probable evidence for milk consumption, and recommended further investigation to clarify the effect of fermented milk, suggesting, ultimately, an increase in the intake of these products as protectors against CRC [81]. A reduction in cancer risk with dairy consumption is frequently associated with high consumption (greater than 250 g/day). Some authors suggest that this is due to the presence of other compounds in milk that also have anticancer properties, such as the aforementioned CLA [82]. Additionally, in the study by Song et al., emphasis was placed on the content of CLA, which has been linked to a lower incidence of CRC, although it is believed that this effect derives from the inverse relationship between whole dairy consumption and the development of CRC [81].

Although the effects of fermented milk are pending corroboration, lactic acid bacteria (LAB) are believed to play an important role in modifying the composition and function of the endogenous microbiota of the host, reducing the absorption of mutagens from cooked foods, inactivating intestinal carcinogenesis, and decreasing intestinal inflammation [81].

A review of meta-analyses aimed at examining the relationship between the intake of different types of dairy products during adulthood and the development of CRC, namely milk, cheese, and fermented milk, by including 15 cohort studies involving approximately 900,000 individuals, 5200 with CRC, revealed a 15% reduction in the risk of CRC associated with the consumption of 500 mL/day of milk, whereas no protective association was observed for the consumption of cheese and fermented milk [85]. Similarly, a meta-analysis of randomized clinical trials did not reveal a reducing effect, probably due to the limited sample size and low event rate (1/668 in the calcium arms and 4/678 in the placebo arms) and the short follow-up periods (range: 3–7 years), which makes achieving significant results difficult [82,86].

In January 2023 [87], an extensive survey was carried out to investigate potential causal relationships between milk consumption and 12 distinct types of cancer. The study leveraged data from white British participants in the UK Biobank (*n* = up to 255,196) and the FinnGen cohort (up to 260,405), as well as information from various cancer consortia. Cancers previously associated with milk consumption in observational studies, along with those prevalent in the UK Biobank and FinnGen populations (>1000 cases), were included. Phenotypic associations of milk intake and cancer incidence were initially assessed in the UK Biobank, followed by a Mendelian randomization (MR) approach to evaluate causality across the UK Biobank, FinnGen consortium, and combined analyses with data from additional consortia for five specific cancers. Meta-analyses were conducted for breast, ovarian, uterine, cervical, prostate, bladder and urinary tract, colorectal, lung, liver, mouth, stomach, and diffuse large B-cell lymphoma, involving cases ranging from 6000 to 148,000 for some cancers. Observational analyses revealed a significant association between milk consumption and an elevated risk of bladder and urinary tract cancer (OR 1.23, 95% CI: 1.03–1.47), with no notable associations observed for other cancer types. However, this association was not substantiated in the Mendelian randomization analysis. Genetically predicted milk consumption, utilizing a variant (rs4988235) near the lactase gene (LCT) locus as a proxy, demonstrated a notable association only with a reduced risk of colorectal cancer (0.89, 0.81–0.98 per additional 50 g/day). According to intercohort Mendelian randomization analyses, genetically predicted milk consumption was linked to a decreased risk of colorectal cancer in the FinnGen cohort (0.85, 0.74–0.97) but was associated with an increased risk of female breast cancer (1.12, 1.03–1.23) and uterine cancer in premenopausal women (3.98, 1.48–10.7) in the UK Biobank. These findings confirm the protective role of milk consumption against colorectal cancer while also indicating potential heightened risks of premenopausal breast cancer and uterine cancer, warranting further investigation.

## 4. Negative Effects of Dairy Consumption

### 4.1. Lactose Intolerance

Lactose is an important source of energy, especially during childhood, and contributes to proper growth and development. As mentioned above, it favors the absorption and use of not only calcium but also other minerals, such as magnesium, zinc, or manganese. It has also been shown to work as a prebiotic, promoting the development of a beneficial intestinal microbiota [54].

Lactose intolerance is a digestive disorder characterized by the inability of the GI tract to digest this milk disaccharide due to the deficiency or absence of β-galactosidase, commonly known as lactase, an enzyme produced in the small intestine and responsible for the hydrolysis of lactose. In the absence of lactose, the lactose reaches the colon intact, where it is consumed by the luminal microbiota, generating waste products such as hydrogen, carbon dioxide, methane, and short-chain acids, such as lactic, acetic, pyruvic, or butyric acids (Figure 2). This fermentation, the result of poor digestion, affects intestinal motility, causing the typical symptoms of lactose intolerance. Its clinical manifestations primarily manifest within the bowel, leading to symptoms such as abdominal pain, bloating, increased flatulence, and diarrhea. Importantly, lactose intolerance is not implicated as a causative factor in instances of rectal bleeding. The amount of lactose needed to induce these symptoms varies among individuals, but it also differs depending on the foods with which it is consumed and the degree of lactase deficiency, so it is not possible to set a single threshold [54].

According to the causes that cause intolerance, we can differentiate four types: primary, secondary, developmental, or congenital. Primary lactase deficiency is the most common variant and is genetically determined. In these cases, the production of β-galactosidase begins as normal, but after two years (although it may be at a more advanced age), it decreases progressively, though symptoms do not appear until adolescence or adulthood. Secondary deficiency is caused by temporary damage to the small intestine, usually by infections or diseases such as viral gastroenteritis, which are quite frequent during childhood after episodes of acute gastroenteritis. Treating the underlying problem usually resolves it, making it transient and recoverable. Developmental deficiency occurs in premature infants as a result of insufficient levels of lactase due to insufficient maturity of the GI tract, which is a temporary situation. Finally, congenital deficiency is an autosomal recessive genetic variant and is characterized by zero or very low lactase production since birth [88].

Most people who suffer from this situation tend to unnecessarily decrease and even eliminate the consumption of milk and milk products, leading to insufficient intake of essential nutrients such as calcium, riboflavin, vitamin B12, or high-quality protein. Before a lactose-free diet is prescribed, such intolerance must be diagnosed and confirmed by a medical professional via the recognized methods. There are dairy products in which the lactose naturally present in raw milk has been reduced and even eliminated owing to the technological process to which it is subjected. Therefore, people with diagnosed lactose intolerance could resort to the consumption of these products, avoiding those products specially prepared for this group, which usually have higher costs. For example, according to the data obtained from USDA Food Data Central (https://fdc.nal.usda.gov/, accessed on 16 December 2024), per 100 g of product, 0.66 g of lactose is present in cheddar cheese. Yogurt is 3.35 g in mass, but its concentration decreases with conservation time, and the bacteria present in the products help in its digestion. Similarly, Malmir et al. reported an inverse association between milk and dairy consumption and the risk of osteoporosis and hip fracture in cross-sectional and case-control studies [89].

### 4.2. Cow Milk Allergy

CMPA is one of the most common food allergies, with a prevalence in developed countries between 0.5% and 3% at the age of 1 year [90]. Among peanuts and tree nuts, CMPA is one of the most common causes of food-induced anaphylaxis [91]. This allergy appears after dairy intake. This occurs mostly in infancy and early childhood, typically in the first 12 months of life. After that, an abnormal immune response, which is mediated by immunoglobulin E (IgE) and not mediated by IgE or mixedis, manifests. IgE-mediated reactions typically manifest promptly following ingestion (from minutes to 1 to 2 h of ingestion) [90], whereas non-IgE-mediated reactions exhibit a delayed onset, requiring up to 48 h to emerge. Despite the deferred timeframe, non-IgE-mediated reactions still engage the immune system. The symptoms associated with non-IgE-mediated conditions are mainly gastrointestinal [90]. For that reason, non-IgE symptoms are frequently misidentified as signs of intolerance, leading to the use of terms such as “lactose intolerance” or “milk intolerance”. However, the symptoms associated with LI (only the intestine, for example, pain, flatulence, and diarrhea) are different from those associated with IgE CMPA (gastrointestinal but also cutaneous or respiratory). Although it varies according to population and age, 60% of cases of cow protein allergy are IgE-mediated [90]. Proper management and diagnosis of milk allergy in children are crucial [88].

Cow milk contains at least 25 different proteins, and allergic people are sensitive to several of them at the same time, the most allergenic caseins being of types α, β, and κ. Serum proteins such as β-lactoglobulin and α-lactalbumin are also important allergens [92]. Milk processing negatively affects several beneficial compounds present in fresh cow milk that have a protective effect on allergies and infections. The main proteins in milk undergo alterations due to industrial processing, which is very important from an immunological point of view. These alterations include denaturation and the presence of new antigenic epitopes. In addition, the destruction of immunomodulatory factors and cytokines promotes the tolerability of milk [93]. Therefore, it is clear that in the innovation of milk treatment, methods that avoid protein alterations would be interesting.

Dietary management involves removing allergenic protein from the diet. Some authors [88] suggest eliminating all dairy products from the nursing mother’s diet if a milk allergy is suspected in the baby, alongside the administration of calcium supplements. However, our previous review indicated that human milk primarily contains fragments from bovine proteins, with potentially allergenic molecules being more prevalent in mothers with allergic tendencies [94]. Despite this, a definitive connection between maternal diet and the allergen content in breast milk could not be firmly established. Infants receiving milk from human milk banks, where donor milk is pasteurized, may face an increased risk of developing allergies, particularly to β-lactoglobulin.

For formula-fed babies, the choice of formula depends on symptom severity. Most infants respond well to extensively hydrolyzed formulas, where milk proteins are broken down. Amino acid formulas are reserved for severe symptoms or if there is no response to extensively hydrolyzed formulas. A first-line amino acid formula is also recommended if complementary foods are necessary for an exclusively breastfed infant showing symptoms suggestive of cow milk allergy [88].

## 5. Inconclusive Evidence on Dairy Consumption

### 5.1. Protection in Type II Diabetes Mellitus

Foods containing digestible carbohydrates increase postprandial blood glucose levels and glycemic responses. This response depends on factors such as the amount and type of carbohydrate, the gastric emptying time, its release from the food matrix, or the rate of absorption. Compared with vegetable drinks, milk may have a protective effect against glucose spikes due to a slower gastric emptying time, resulting in a sustained glucose and galactose absorption rate due to the need for enzymatic hydrolysis in the intestine and stimulated insulin secretion [95]. However, the protective effect of milk in type II diabetes mellitus is still controversial. A recent review of several studies revealed that there is an inverse relationship between high milk fat consumption and the development of type II diabetes and that dairy may play a protective role. Other studies have shown that whole yogurt intake is a preventive factor but not for skimmed yogurt, whole and skimmed milk, or butter [96]. In contrast, the findings from a prospective study involving 3454 nondiabetic individuals from the Spanish study PREDIMED reflect that the protective role of dairy products is mainly due to low-fat dairy products, as an increase in the consumption of skimmed dairy products and yogurt in general during the follow-up period was inversely associated with the onset of type II diabetes. It was even proven that the exchange of daily rations of biscuits, chocolate, wholemeal biscuits, or homemade pastries for a daily serving of yogurt supposes a risk reduction of 40–45% [97]. Undoubtedly, the data are still highly contradictory. There are many factors that may influence the differences between studies, including the population studied and how it was selected and the technique used to quantify the type and amount of dairy. Although the questionnaires used are usually extensively validated, unavoidable errors can always occur [98]. There are also other variables that can influence the effects of dairy products, such as lifestyle or other dietary components.

### 5.2. Contributing to Weight Gain

It is very common to hear the expression “milk fattening”; however, scientific evidence indicates the opposite, or in some cases, it simply states that it has no effect. Despite these studies suggesting a beneficial association between dairy consumption and weight loss or modification of body composition, the results of clinical trials are inconclusive. On the one hand, several reviews of clinical trials agree that an increase in dairy consumption, mainly in the form of skimmed dairy products, yogurt in general, and skim milk, has no significant effect on weight loss if caloric restriction is not applied simultaneously. In conjunction with caloric restriction, positive effects of dairy on weight loss and body fat have been observed [99,100]. In other cases, it is argued that the favorable effect on weight reduction is not due to a higher consumption of low-fat dairy products but to the healthier eating pattern that accompanies this habit. That is, the substitution of questionable nutritional quality foods (sugary drinks, sweets, etc.) with dairy products [100,101,102].

A series of mechanisms have been postulated to explain the effects of dairy products on body weight and fat. On the one hand, an increase in the intake of these foods leads to an increase in calcium intake, which favors weight loss since elevated calcium levels reduce lipogenesis and stimulate lipolysis. Moreover, this mineral creates insoluble complexes with fatty acids in the gut, limiting the absorption of dietary fats. Other dairy components, such as whey proteins, which contribute to muscle maintenance and promote lipid metabolism, have also been proposed as weight loss promoters. Another example is CLA, which is also involved in the regulation of lipid metabolism, adipogenesis, and inflammation. In the case of yogurt, its relationship with a lower risk of developing overweight or obesity is also attributed to the probiotics it contains, which act as modulators of the inflammatory balance, preventing weight gain [100]. Although most studies recognize the beneficial role of dairy products, the evidence is still inconclusive [101].

### 5.3. Impaired Lipid Profile and Increased Risk of Cardiovascular Disease (CVD)

Milk is one of the main sources of saturated fat in the diet. Lauric acid, myristic acid, and palmitic acid account for a significant percentage of the total fatty acids in milk. Its elimination is common as a preventive measure against CVD, since hypercholesterolemic properties have been attributed to this elimination in the case of excessive consumption. However, evidence of the relationship between dairy consumption and increased mortality due to this pathology does not include dairy restriction as an effective preventive factor [103].

Although scientific evidence has linked the consumption of saturated fat with increased LDL cholesterol (LDL-c) levels and the risk of developing CVD, recent studies have revealed that the relationship between saturated fat and CVD could be less direct than initially thought. This is because some foods high in saturated fat contain both saturated and unsaturated fatty acids, each of which affects the lipoproteins involved in metabolism differently. In a review of observational studies, the majority reported that dairy intake was associated with an increased risk of CVD, regardless of fat percentage. Other studies reported that a diet high in saturated fat from whole milk and butter increased LDL-c when carbohydrates or unsaturated fatty acids were replaced. However, they also increase HDL cholesterol (HDL-c), so it does not increase, or even reduces, the total cholesterol/HDL-c ratio [104]. A study that included data from 0.9 million individuals from the China Kadoorie Biobank and the United Kingdom biobank reported that total dairy consumption reduces the risk of CVD by 3.7% and the risk of stroke by 6%, respectively. Additionally, the consumption of cheese and low-fat dairy products is inversely associated with CVD [105]. The results of this study are quite relevant because they are from two different parts of the world.

Another recent study that included 4438 Japanese patients with type II diabetes mellitus associated yogurt and milk with cardiometabolic risk factors. A high consumption of milk and yogurt was associated with a lower body mass, blood pressure, serum triglyceride concentration, and urine albumin/creatinine ratio. In addition, the consumption of these dairy products is also associated with a lower probability of suffering from metabolic syndrome and chronic kidney disease [106]. Importantly, high consumers of dairy products had higher levels of LDL. Although both low-fat and high-fat yogurt and milk decreased the risk of metabolic syndrome, this association was statistically significant only for high-fat products. Therefore, milk fat could play a protective role.

### 5.4. Risk Factors for Prostate Cancer Development

There is growing evidence that dairy-rich diets constitute a risk factor for the development of prostate cancer. There is a probable association between calcium-rich diets and an increased risk of prostate cancer. However, the evidence is limited for the increased risk associated with milk and dairy consumption [107,108]. A systematic review and meta-analysis reviewed 32 cohort studies and revealed that high intakes of dairy products (400 g/day), milk (200 g/day), skim milk (200 g/day), cheese (50 g/day), and calcium (400 mg/day) increase the total risk of prostate cancer. Similar results were reported by Steck et al. [109], who reported that higher Ca:Mg ratios and whole milk intake were associated with more aggressive forms of prostate cancer. However, a case-control study in Australian men tested the relationships of diet and body size with prostate cancer risk and concluded that the intake of milk and dairy products was inversely associated with prostate cancer risk [110]. In a recent meta-analysis, Zhao et al. reported differences between the types of dairy products. Total dairy milk, cheese, and butter consumption increased the risk of prostate cancer. However, the consumption of whole milk reduces the risk of prostate cancer [111]. The authors hypothesize that the association between dairy consumption and prostate cancer could be due to increased intake and blood circulation of IGF-1, a molecule that has been associated with the development of this type of cancer [112].

### 5.5. Autism Spectrum Disorder (ASD)

Autism spectrum disorder (ASD) involves a neuronal dysfunction that affects socialization, communication, imagination, planning and emotional reciprocity, and evidence of repetitive or unusual behaviors. The prevalence of ASD has increased steadily over the last few decades [113]. Although GI dysfunction in the form of chronic constipation, diarrhea, or abdominal pain is common among children with ASD, it is usually ignored or not recorded due to communication difficulties. The evidence suggests that in these patients, the GI structure is altered, and their intestinal permeability is increased by 25.60%, compared with 2.30% in healthy children [114]. This, together with the enzymatic deficiency they present, contributes to the development of abnormal immune reactions against food components. Interestingly, patients with ASD who follow a protein-restricted diet, specifically wheat and dairy-free diets, have significantly less permeability and an improvement in the associated neurological symptoms [114,115]. The mechanism by which these components of the diet, especially dairy proteins, aggravate the symptoms of ASD is unclear. However, it has been hypothesized that, after incomplete digestion of milk proteins, specifically β-caseins, opioid peptides called β-casomorphine 7 (β-CM7) are generated (Figure 3). These peptides can pass into the bloodstream owing to their high intestinal permeability, cross the blood–brain barrier, and interact with µ-opioid receptors involved in regulating aspects of social behavior. This binding alters neurotransmission patterns and increases the activity of the endogenous opioid system, which is related to the pathogenesis of ASD [114,116,117]. Simultaneously, the presence of β-CM7 in the intestinal lumen decreases the absorption of the amino acid cysteine (CYS), a necessary precursor for the synthesis of glutathione (GSH) tripeptide. The oxidative stress associated with low levels of GSH enhances the release of cytokines, contributing to inflammation of the GI tract and leading to GI discomfort and dysfunction, which are characteristic of patients with ASD [114].

The predominant variants of β-casein in cow milk are A1, A2, and B, with types A1 and B being those from which β-CM7 is generated [118]. The presence of one or another variant is genetically conditional; milk from Holstein–Friesianis contains a mix of A1 and A2 variants, whereas the A2 variant predominates in the Guernsey and Jersey breeds. Therefore, genetic selection in livestock may be a possible way to guarantee the absence of β-casein A1 and ensure that patients with ASD do not have to resort to dairy-restricted diets [114,119,120,121].

Despite the strength of the experimental data, there is controversy about the real clinical effects on ASD patients, since the results are contradictory and the methods used are varied and difficult to compare, so more rigorous research is necessary to achieve solid scientific evidence.

### 5.6. Source of miRNAs

MicroRNAs (miRNAs), noncoding RNA molecules ranging from 21–25 nucleotides in length, are capable of binding to the untranslated regions (UTRs) of the 3′ ends of messenger RNA (m-RNA), which makes them responsible for 40–60% of the regulation of genetic expression at the posttranscriptional level [122,123]. These molecules have been identified in various biological fluids, including milk [124,125]; in other words, they are transferred between individuals, between mothers and children, during breastfeeding. This paradigm can be extended to the micro-RNAs of bovine milk consumed by the adult population, since there is strong evidence both of their presence and of the fact that the human species absorbs them in quantitative and biologically significant quantities [126,127,128], presenting the majority of immunological properties and the ability to alter gene expression in different tissues [122,123,129,130].

Most studies on the bioactivity of milk micro-RNAs have focused on micro-RNA 148a, one of the most abundant in both human and bovine milk [123,131]; that is, it is entirely preserved among human and bovine species, thus acting on the same targets as its human counterpart. This microRNA is involved in several processes, including development, adipogenesis, appetite control, immunity, cell proliferation, and homeostasis [123].

One of its corroborated effects is the repression of DNA methyltransferase 1 (DNMT1), an enzyme inhibitor of gene expression, by the suppression of its transcription. The regulation of this enzyme plays an essential role in the development and control of mammalian growth, as it is responsible for the methylation of DNA cytosine in the CpG islands, which are DNA regions that encode approximately 70% of gene promoters in humans and intervene in genetic silencing. Its inhibition results in less methylation of developmental-related gene promoters in the CpG islands, such as FOXP3 (regulatory T-cell master transcription factor), insulin (INS), insulin-linked growth factor 1 (IGF-1), or fat mass- and obesity-associated gene (FTO), and, consequently, greater expression of the latter [123,131,132]. The overexpression of these genes, together with the interaction of microRNA 148a with other targets, enhances mechanisms that promote intake, adipocyte deposits, and weight gain, inhibits satiety mechanisms, and interferes with diabetogenic processes [131,132].

Like microRNA (microRNA) 148a, microRNA 200c is conserved between human and bovine species and is one of the 10 most abundant microRNAs in the milk of both species [128,133]. It participates in several processes associated with the development and progression of cancer, such as the preservation of stem cell characteristics, alteration of the regulation of programmed death, and chemosensitivity [134]. Its more documented functions are related to the suppression of epithelial-mesenchymal transition (EMT), one of the first steps in the development of metastasis that favors the motility of tumor cells [125,127,134,135]. In other words, microRNA 200c is capable of inducing mechanisms that suppress the progression of cancer.

Other less abundant micro-RNAs, with abundant bibliographies behind them, are those belonging to the micro-RNA 29 family, which is composed of three members, 29a, 29b, and 29c, all of which are involved in various diseases. In the case of milk, the presence of microRNA 29b in the blood serum of consumers is dose-dependent [136,137]. Specifically, this microRNA has been associated with different mechanisms related to cancer, such as apoptosis, cell proliferation, or EMT, and inhibits the expression of oncogenes and promotes the expression of tumor suppressors [138,139,140]. Low levels of microRNA 29b have been associated with more aggressive forms of the disease, suggesting that re-establishing them may lead to improved prognosis [139,140].

There are few studies aimed at examining the role of microRNAs in the homeostasis of skeletal tissue; however, most agree with the role of microRNA 29b as a promoter of bone mineralization, favoring the differentiation of osteoblasts and suppressing that of osteoclasts [141,142]. In contrast to the potential benefits derived from the intake of microRNA 29b through bovine milk, similar to microRNA 148a, it can implement certain mechanisms related to the development of type II diabetes mellitus [126,132,143].

Although some of the effects of milk microRNAs present strong scientific evidence, there is still a great lack of knowledge about the totality of their functions, so it is not possible to determine to what extent they may be beneficial or detrimental to the health of consumers.

## 6. Milk vs. Plant-Based Drinks: The Role of the Milk Fat Globule Membrane

Currently, the consumption of vegetable drinks as milk substitutes is increasing. This is due not only to lactose intolerance but also to various ethical or cultural reasons. The nutritional characteristics of both products are clearly different (Table 5). For example, milk contains more energy, fats, carbohydrates, vitamins C, B2, B12, and A, biotin, pantothenic acid, calcium, and phosphorus than does soy milk or other vegetable drinks. In addition, with the exception of soy, the protein content is considerably higher in milk than in other vegetable drinks since these beverages have a composition of less than 1% protein. In addition, milk has a relatively high score for essential digestible amino acids [144]. However, undoubtedly, one of the main differences is the absence of the milk fat globule membrane (MFGM) in vegetable drinks. This compound is considered to have great potential as a nutraceutical, and MFGM supplementation is gaining increasing attention [145].

This complex structure has a thickness of 10–50 nm [146], is characteristic of mammals, and is secreted by milk-producing cells during lactation. The MFGM is composed of three layers surrounding milk fat globules and represents a mixture of lipids and proteins with a fundamental role in the stability and functionality of this structure. Among the lipids, MFGM is mainly composed of polar lipids such as glycerophospholipids, which include phosphatidylcholine (PC, 31–35%), phosphatidylethanolamine (PE, 30%), phosphatidyly nositol (PI, 5–7%), phosphatidylserine (PS, 3–5%), sphingolipids, gangliosides, cerebrosides (lactosyl-cerebroside (LacCer; 3.4%), and glucosyl-cerebroside (GlucCer; 0.3%), as well as ceramides and sphingosines [147,148]. Among the MFGM proteins are mucins, xanthine dehydrogenase/xanthine oxidase, butyrophilin, lactoferrin, lactadherin, adipophilin (ADPH), and fatty acid binding protein (FABP) [147,149].

MFGM has been shown to have various positive effects on health, but undoubtedly, one of the most important points is its effect on the neuronal development of children. Various studies have evaluated this effect. For example, Colombo et al. [150] evaluated the effect on the neurological development of children who received conventional infant formula versus those who received formula that also contained MFGM and lactoferrin. The results revealed that children fed for up to 12 months with infant formula supplemented with MFGM and lactoferrin presented better cognitive results at 5 and a half years of age in multiple domains, including measures of intelligence and executive function. Similarly, at one year of study, children supplemented with MFGM and lactoferrin presented accelerated neurodevelopment and fewer medical complications than did the control group [151]. However, the positive effects are not limited to children; positive effects have also been shown in adults. For example, adults who followed a diet enriched in MFGM scored lower in stress tests than those in the placebo group did, which could suggest an ability to reduce anxiety and improve general psychological health [152]. Another study evaluated the effect of consuming MFGM-enriched milk in subjects over 65 years of age. After 14 weeks of study, participants showed an improvement in episodic memory, especially women [153].

The consumption of dairy products, due to the presence of MFGM compared with the consumption of vegetable drinks, may present other diverse benefits. MFGM can promote cholesterol homeostasis and metabolic health, skeletal health, and motor skills, or intestinal ecology and immunity [147]. A recent review [154] compared the effects of milk consumption and vegetable drink consumption on the intestinal microbiota. The results showed that milk consumption increased the diversity and richness of the microbiota and the abundance of beneficial microorganisms such as *Bifidobacterium*, Lactobacillus, *Akkermansia*, *Lachnospiraceae*, and *Blautia*. Some of these positive effects on the microbiota have been linked to the presence of MFGM. For example, Li et al. [155] reported that MFGM supplementation increased the Bacteroidetes/Firmicutes ratio in rats fed a high-fat diet. Likewise, it has been observed that the joint supplementation of MFGM with probiotics can have a synergistic effect [156].

## 7. Conclusions

Cow milk is undoubtedly one of the most complex foods from a nutritional and consumer point of view. The adverse health effects and motivations, such as animal welfare or the environment, are some of the conditions that this product faces. The nutritional value of its high-biological-value protein, its fatty acids, or its levels of vitamins or minerals is fully understood. However, dairy products are an important source of bioactive peptides, prebiotics, and beneficial bacteria and can help prevent osteoporosis or other types of diseases. Another component not present in other foods, MFGM, plays a key role in lipid metabolism and neuronal development. Allergies to protein or lactose intolerance are the main problems related to its consumption. However, one of the factors that creates the most controversy regarding dairy consumption is its effect on cardiovascular health, weight and type II diabetes. Although most meta-analyses have shown contradictory results, recent studies have shown that dairy consumption has a beneficial cardiovascular effect. Even so, factors such as the influence of fat content are still a matter of debate. Another factor that will be interesting to evaluate in the coming years is whether milk microRNAs play an important role in consumer health and from what levels of consumption. These types of studies will be key to elucidating the real role of milk in consumer health. The evidence obtained will also be highly important when developing dietary recommendations for different age ranges.

## Figures and Tables

**Figure 1 nutrients-17-00646-f001:**
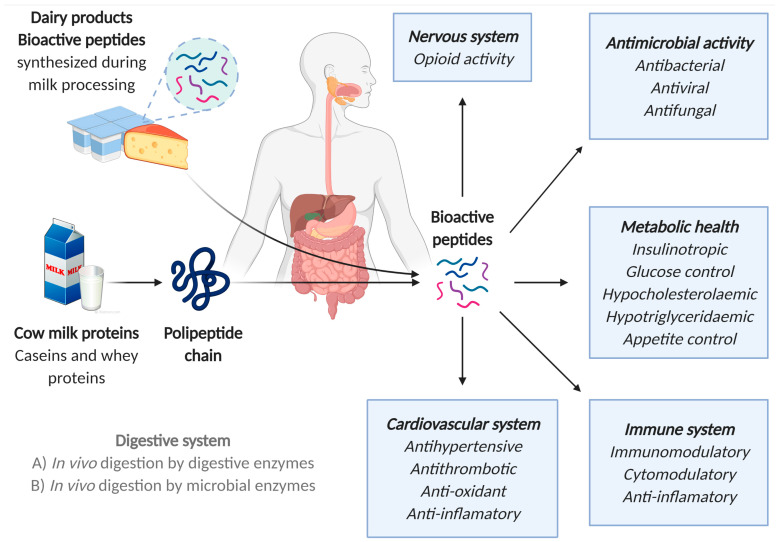
Schematic figure of the in vivo generation of bioactive peptides from milk proteins and their physiological functions. Figure based on [48]. Created with Biorender.com.

**Figure 2 nutrients-17-00646-f002:**
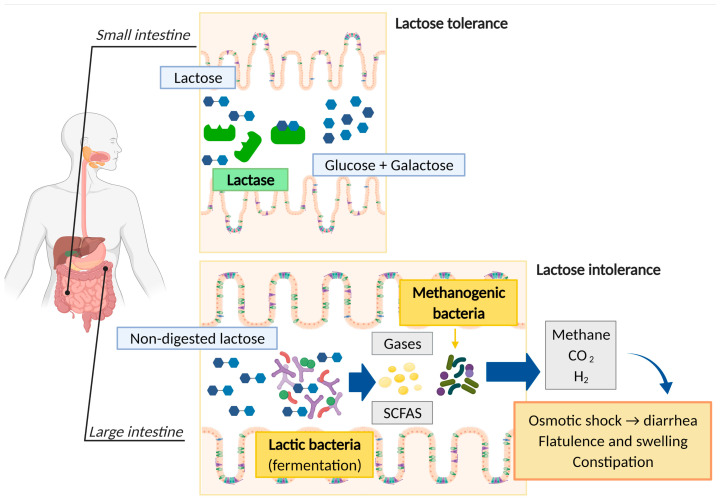
Schematic figure of the routes followed by lactose in cases of tolerance and lactose intolerance. Created with Biorender.com.

**Figure 3 nutrients-17-00646-f003:**
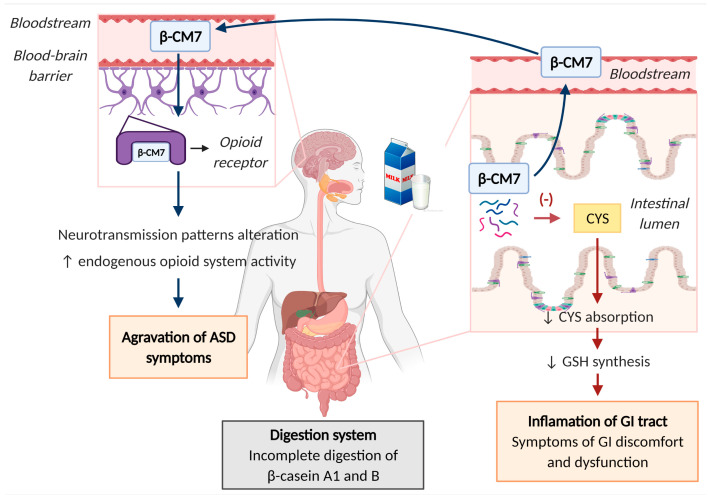
Schematic figure of the mechanism by which β-CM7 from dairy β-casein A1 and B aggravates the symptoms of ASD. Created with Biorender.com.

**Table 1 nutrients-17-00646-t001:** Nutritional composition of cow milk and dairy products: energy, macronutrients, and micronutrients (per 100 g of edible portion). Data adapted from USDA FoodData Central (https://fdc.nal.usda.gov/, accessed on 10 December 2024) and [7].

Nutrient	Whole Milk	Semi-Skimmed Milk	Skim Milk	Whole Natural Yogurt	Whole Greek Yogurt	Skimmed Natural Yogurt	Burgos Cheese	Cured Manchego Cheese	Cheddar Cheese	Swiss Cheese	Ricotta Cheese	Parmesan Cheese
Energy (Kcal)	65.4/274	47.6/199	37/155	61.4/257	95	44.9/188	200/837	476/1992	409	393	158	420
Proteins (g)	3.10	3.50	3.90	4	8.78	4.30	14	38	23.30	27	7.81	29.60
Total lipids (g)	3.80	1.60	0.20	2.60	4.39	0.32	14.90	35.80	34	31	11	28
Saturated fatty acids (g)	2.30	1.10	0.09	1.50	2.39	0.11	8.80	18.70	19.2	18.2	6.97	15.5
Monosaturated fatty acids (g)	1.10	0.45	0.06	0.72	0.96	0.15	4.30	8.40	7.44	7.26	2.56	6.40
Polyunsaturated fatty acids (g)	0.13	0.04	0.01	0.13	0.11	0	0.89	6.20	1.18	1.14	0.39	1.20
Cholesterol (g)	14	6.30	2.60	10.20	17	1	14.50	74.40	100	93	48	87
Carbohydrates (g)	4.70	4.80	4.90	5.50	4.75	6.30	2.50	0.51	2.44	1.44	6.86	12.40
Water (g)	88.40	90.10	91	87.90	81.30	89.10	68.60	25.70	36.60	37.60	72.90	22.80
Calcium (mg)	124	125	121	142	111	140	191	848	707	890	224	884
Iron (mg)	0.09	0.09	0.09	0.09	<0.10	0.09	0.62	0.75	0.16	0.13	0.10	0.45
Iodine (µg)	9	8.60	11.10	3.70	42.30	5.30	4.80	34	N.D	N.D	N.D	N.D
Magnesium (mg)	11.60	11.90	28.60	14.30	10.70	13.70	24.40	33.50	26.80	33.40	19.70	34.90
Zinc (mg)	0.38	0.52	0.54	0.59	0.47	0.44	2	3.20	3.67	4.37	0.56	4.33
Sodium (mg)	48	47	53	80	34	57	294	742	654	185	105	1750
Potassium (mg)	157	155	150	280	147	187	200	100	77	71	230	184
Phosphorus (mg)	92	91	97	170	126	109	600	560	458	574	162	634
Selenium (µg)	1.40	1.50	1.60	2	N.D	1	14.50	1.60	28.30	30.1	5.50	35
Thiamine (mg)	0.04	0.04	0.04	0.04	0.06	0.04	0.02	0.04	0.03	0.01	0.04	0.03
Riboflavin (mg)	0.19	0.19	0.17	0.18	0.24	0.19	0.17	0.33	0.44	0.30	0.33	0.35
Niacin equivalents (mg)	0.73	0.71	0.90	0.44	0.23	1.20	4.10	7.20	0.05	0.06	0.17	0.08
Vitamin B_6_ (mg)	0.04	0.06	0.04	0.05	0.04	0.08	0.08	0.20	0.07	0.07	0.10	0.08
Folate (µg)	5.50	2.70	5.30	3.70	N.D	4.70	14.30	21.80	21	9	4	6
Vitamin B_12_ (µg)	0.30	0.30	0.30	0.20	N.D	0.40	0.66	1.50	1.06	3.02	0.78	1.35
Vitamin C (mg)	1.40	0.52	1.70	0.70	N.D	1.60	0	0	N.D	N.D	N.D	N.D
Vitamin A (retinol equivalents, µg)	46	18.90	Traces	9.80	38	0.80	261	234	316	292	127	228
Vitamin D (µg)	0.03	0.02	Traces	0.06	<0.01	0	0	0.19		0	N.D	N.D
Vitamin E (mg)	0.1	0.04	Traces	0.05	N.D	Traces	0.56	0.61	0.75	0.6	0.27	0.51

N.D: Not determined.

**Table 2 nutrients-17-00646-t002:** Comparison of essential amino acids in cow milk and protein patterns in children aged 1–3 years. Data adapted from [7].

Amino Acids	Protein Consumption Pattern (mg/g)	Milk Protein (mg/g)
Histidine	18	25
Isoleucine	25	56
Leucine	55	92
Lysine	51	72
Methionine + Cysteine	25	30
Phenylalanine + Tyrosine	47	56
Threonine	27	41
Tryptophan	7	13
Valine	32	62

**Table 3 nutrients-17-00646-t003:** Milk composition (% of total fatty acids) in terms of the major fatty acids of cow milk fat [13,19].

Fatty Acid	% of Total Fatty Acids
Butyric 4:0	3.30
Capric 10:0	2.60
Caprylic 8:0	1.20
Caproic 6:0	1.90
Stearic 18:0	11.00
Lauric 12:0	3.50
Linoleic 18:2	2.40
α–linolenic 18:3	0.50
Myristic 14:0	11.50
Oleic 18:1	28.00
Palmitic 16:0	25.80
Palmitoleic 16:1	2.90

**Table 4 nutrients-17-00646-t004:** Comparison of absorbable calcium from different dietary sources [44].

Food	Ration Size (g)	Ca Content (mg)	Fractional Absorption	Estimated Absorbable Ca (mg)	Rations Equivalent to One Serving of Milk
Milk	240	300	32.10	96.30	1
Yogurt	240	300	32.10	96.30	1
Cheddar cheese	42	303	32.10	97.20	1
Pinto bean	86	44.70	26.70	11.90	8.10
Red bean	172	40.50	24.40	9.90	9.70
White bean	110	113	21.80	24.70	3.90
Broccoli	71	35	61.30	21.50	4.50
Kale	85	61	49.30	30.10	3.20
Spinach	85	115	5.10	5.90	16.30
Sweet potato	164	44	22.20	9.80	9.80
Tofu with calcium	126	258	31	80	1.20

**Table 5 nutrients-17-00646-t005:** Macronutrient composition of milk and vegetable drinks on the basis of data from FoodData Central (https://fdc.nal.usda.gov/, accessed on 16 December 2024).

Nutrient	Whole Milk	Soy-Based Drink	Oat-Based Drink	Almond-Based Drink
Energy (Kcal)	274	38	48	15
Proteins (g)	3.10	3.55	0.80	0.55
Total lipids (g)	3.80	2.12	2.75	1.22
Saturated fatty acids (g)	2.30	0.31	N.D	0.10
Monosaturated fatty acids (g)	1.10	0.42	N.D	0.73
Polyunsaturated fatty acids (g)	0.13	1.15	N.D	0.28
Cholesterol (g)	14	-	-	-
Carbohydrates (g)	4.70	1.29	5.10	0.34
Water (g)	88.40	92.40	90.60	97.40

N.D: Not determined.

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
