# Peer review of "The Role of Dairy in Human Nutrition: Myths and Realities"

_nutrients, 2025, doi:10.3390/nu17040646_

Round 1

Reviewer 1 Report

Comments and Suggestions for Authors

The review titled “The role of milk on human nutrition: myths and realities” aims to explore key issues of the nutritional composition of milk, offering insights into its real role in promoting human health. It seeks to dispel myths and contribute to a clearer understanding of the intricate relationship between milk consumption and human well-being.

The paper starts with the milk nutritional composition referring on limited references (Spain). It would be expected to be a table with more international sources and data. It is suggested to add the word “cow” because the literature cited, and the case-studies deal with cow milk. Moreover, it is suggested to consider dairy products as well.

Other comments are included in the pdf attached.

Comments on the Quality of English Language

In general, the text needs a deep improvement as far as language (for example, many times “like” instead of “such as”) together with misspelling issues (see the pdf attached).

Author Response

The review titled “The role of milk on human nutrition: myths and realities” aims to explore key issues of the nutritional composition of milk, offering insights into its real role in promoting human health. It seeks to dispel myths and contribute to a clearer understanding of the intricate relationship between milk consumption and human well-being.
Comment 1: The paper starts with the milk nutritional composition referring on limited references (Spain). It would be expected to be a table with more international sources and data. It is suggested to add the word “cow” because the literature cited, and the case-studies deal with cow milk. Moreover, it is suggested to consider dairy products as well.
Response 1: Many thanks for the comment. We have added more dairy products in Table by using the data present in USDA FoodData Central (https://fdc.nal.usda.gov/). We also included in the title in the Table the word cow to indicate that these products are elaborated with cow milk.
Comment 2. Line 3: the keywords include cow dairy and fermented milk. Change the title accordingly to the issues of the paper
Response 2: We have changed Milk by dairy in the title. 
Comment 3. Line 23: cow milk, because the literature reported deals with this milk
Response 3: Modified. 

Reviewer 2 Report

Comments and Suggestions for Authors

This review article evaluates the role of milk and dairy products in the human diet, highlighting their benefits and potential negative effects.

There are several issues that should be considered to enhance the quality of the review.

The abstract is insufficiently developed and lacks scientific rigor.

Lines 26-28, In the Codex Alimentarius [1], milk is defined as “the normal mammary secretion of dairy animals obtained by one or more milking’s without any type of addition or extraction, destined for consumption in the form of liquid milk or further processing”. The reference comes after the statement not before it.

Lines 33-37, which reference said “ Despite its evident importance…….web-circulating in recent years.” I do not think this information is accurate.

Line 43, what does LI stand for?

Line 113, “Although the levels of this amino acid in cow's milk are…”  You are talking about fatty acids, not amino acids, so change it.

Is the Cheddar cheese Ration size correct in Table 4.?

 Figure 1, where are A and B located in the schematic?

The rationale for including this section, “6. Milk vs plant-based drinks: the role of milk fat globule membrane,” is unclear.

Comments on the Quality of English Language

Proofreading is required because there are a lot of incomplete sentences and some other writing issues.

Author Response

Reviewer 2
This review article evaluates the role of milk and dairy products in the human diet, highlighting their benefits and potential negative effects.
There are several issues that should be considered to enhance the quality of the review.
Comment 1: The abstract is insufficiently developed and lacks scientific rigor.
Response 1: The abstract was modified. 
Comment 2: Lines 26-28, In the Codex Alimentarius [1], milk is defined as “the normal mammary secretion of dairy animals obtained by one or more milking’s without any type of addition or extraction, destined for consumption in the form of liquid milk or further processing”. The reference comes after the statement not before it.
Response 2: Modified.
Comment 3: Lines 33-37, which reference said “ Despite its evident importance…….web-circulating in recent years.” I do not think this information is accurate.
Response 3: Many thanks for the comment we have modified the paragraph to be more accurate. 
“Despite its evident importance, over the last 20 years the consumption of milk has declined in Western European countries and US [3]. This is because consumers have begun to question milk as a perfect food due to a possible association with health risks, taste or animal welfare and environmental protection reasons [3]. Curiously, the consumption of dairy products such as cheese remains stable or increases, which may be due, among other things, to the fact that there are no plant-based alternatives as established as in the case of milk [3]. Among health problems, many people associate digestive problems with milk consumption. However, most people who stop consuming this food for this reason do so without consulting a doctor or undergoing a lactose intolerance test [4]”
Comment 4: Line 43, what does LI stand for?
Response 4: Lactose intolerance. Included in the manuscript 
Comment 5: Line 113, “Although the levels of this amino acid in cow's milk are…”  You are talking about fatty acids, not amino acids, so change it.
Response 5: Modified.
Comment 6: Is the Cheddar cheese Ration size correct in Table 4.?
Response 6: Yes in Ration size mentioned in the original reference. 
Comment 7: Figure 1, where are A and B located in the schematic?
Response 7: In this case A and B was used to list the both alternatives not refers to any other part of the figure. 
Comment 8: The rationale for including this section, “6. Milk vs plant-based drinks: the role of milk fat globule membrane,” is unclear.
Response 8: This section was included based on the comments of the Editor. 

Reviewer 3 Report

Comments and Suggestions for Authors

This manuscript presents a comprehensive narrative review of the role of milk and dairy products in human nutrition. The aim is to deal with the complexity of milk consumption, both its nutritional advantages and the controversy surrounding its effects on health. The overview examines the multifaceted composition of milk, including macronutrients, micronutrients, and bioactive components. It evaluates their effects on human health, whereby both the positive and the negative aspects of milk consumption are considered. It also examines the role of milk in various health conditions, such as osteoporosis, cancer, type-II diabetes, and autism spectrum disorder. In addition, the manuscript milk contrasts with vegetable alternatives and focuses on the unique milk components such as the milk fat ball membrane (MFGM). As a narrative review, there may be statistical strictness in the meta-analysis. While the authors strive for objectivity, narrative ratings can be susceptible to distortions of authors.

·         Abstract – Consider adding further details about the advantages. Strengthen the role of scientific evidence. I suggest a more detailed presentation of the controversy. The role of fermented dairy products is not mentioned in the summary. The abstract mentions the practical implications, but it is worth clarifying what this means in the context of nutritional recommendations.

·         Introduction – The topic of pseudo-scientific articles and widespread misunderstandings in recent years should be expanded. Specific examples of such false beliefs found online should be provided. In the introduction, the various mechanisms and dietary effects could be mentioned in each disease. In addition, the introduction could briefly respond to the potential advantages of milk consumption.

·         Nutritional composition of milk and dairy products – It is important to note that the composition of milk can vary depending on numerous factors. The section could contain a more detailed description of the individual types of milk proteins, including information on their functions and their biological meaning. Different types of milk, not just cow's milk, should be considered. The section on milk fats could be expanded to provide more detailed information on the different types of fatty acids and their effects on health. The importance of trans-access acid (TVA) and its potential health benefits should also be emphasized. What about phospholipid content? In addition to lactose, the topic of milk oligosaccharides could be further developed. The mineral section could add further information on how milk contributes to fulfilling the daily requirements for certain micronutrients.

·         Positive effects of dairy consumption – It is important to emphasize that the positive effects of milk are not solely due to individual components but are due to the synergistic effects of all their components. The section could be expanded to include a more detailed discussion of the mechanisms of impact bioactive peptides. The section on prebiotics and probiotics could be deeper in terms of the effects of milk oligosaccharides on the intestinal microbiota. It would also be advantageous to use the effects of fermented milk products on the microbiota and the potential health benefits in connection with their consumption. More context is required for CLA. This section could provide a more detailed description of the protective mechanisms against osteoporosis. The cancer section could be more specific in presenting the mechanisms through which milk and dairy products can exert protective effects.

·         Negative effects of dairy consumption – It is important to note that reactions to dairy products from person to person can vary, depending on factors such as genetics, general health, intestinal bacteria, and nutrition. The type of dairy product is also important. This section could be more detailed on different types of lactose intolerances and their introduction. I suggest a more detailed discussion about the cow's milk protein allergy (CMPA). The section on the possible connection between milk and cancer could be more precise. Consider expanding the topic of the effects on the cardiovascular system. The section on the connection between milk and ASD could be more detailed to explain how beta-casomorphine-7 (beta-cm7), which comes from milk proteins, can influence ASD symptoms. And what about the excess sodium or phosphorus in certain dairy products? What about cholesterol and saturated fat content?

·         Inconclusive evidence on dairy consumption – It should be noted that reactions to milk consumption can be very individual and depend on many factors, such as genetics, state of health, intestinal microbiota, and general nutrition. In the section, the discrepancies in research results regarding the effects of dairy products on the development of type 2 diabetes could be discussed in more detail. The section on the effects of dairy products on weight could explain the different mechanisms that can influence body mass. The section on the effects of dairy products on the lipid profile and the risk of cardiovascular diseases could be more precise. It should also be noted that most evidence does not support the connection between milk consumption and an increased risk of cardiovascular diseases. The section on the connection between dairy products and prostate cancer risk could be more precise when presenting the contradictory research results. It would be worthwhile to discuss the potential causes of these discrepancies. The section on the association of dairy products with ASD could be expanded to provide more detailed information about the potential mechanism of activity of β-casomorphine-7 (β-CM7) for brain function in people with ASD. It should also be noted that not all people with ASD react negatively to dairy products. I believe there is no tabular summary of the data and evidence discussed.

·         Milk vs plant-based drinks: the role of milk fat globule membrane – The section could expand the comparison of the nutritional composition of milk and herbal drinks. It is worth noting that there are many types of vegetable drinks and that their nutritional profiles can vary significantly. In my opinion, a tabular summary of the data discussed is missing. It should be emphasized that the effects of nutrients often depend on the food matrix in which they occur, not only on their individual content. Milk is a complex matrix that can contribute to better bioavailability of certain nutrients. The description of MFGM could be more detailed. The section could discuss the documented health benefits of MFGM in more detail. A discussion of the possible disadvantages of MFGM is just as important.

·         Conclusions – A summary of the advantages of milk and milk product consumption could be more detailed. This section should emphasize more clearly that not all aspects of the effects of milk are clear on health. It would be valuable to add a few words about the effects of the conclusions of the manuscript for nutritional recommendations. The call for further research could be more precise.

Author Response

Comment 1:
This manuscript presents a comprehensive narrative review of the role of milk and dairy products in human nutrition. The aim is to deal with the complexity of milk consumption, both its nutritional advantages and the controversy surrounding its effects on health. The overview examines the multifaceted composition of milk, including macronutrients, micronutrients, and bioactive components. It evaluates their effects on human health, whereby both the positive and the negative aspects of milk consumption are considered. It also examines the role of milk in various health conditions, such as osteoporosis, cancer, type-II diabetes, and autism spectrum disorder. In addition, the manuscript milk contrasts with vegetable alternatives and focuses on the unique milk components such as the milk fat ball membrane (MFGM). As a narrative review, there may be statistical strictness in the meta-analysis. While the authors strive for objectivity, narrative ratings can be susceptible to distortions of authors.
Abstract – Consider adding further details about the advantages. Strengthen the role of scientific evidence. I suggest a more detailed presentation of the controversy. The role of fermented dairy products is not mentioned in the summary. The abstract mentions the practical implications, but it is worth clarifying what this means in the context of nutritional recommendations.
Introduction – The topic of pseudo-scientific articles and widespread misunderstandings in recent years should be expanded. Specific examples of such false beliefs found online should be provided. In the introduction, the various mechanisms and dietary effects could be mentioned in each disease. In addition, the introduction could briefly respond to the potential advantages of milk consumption.
Nutritional composition of milk and dairy products – It is important to note that the composition of milk can vary depending on numerous factors. The section could contain a more detailed description of the individual types of milk proteins, including information on their functions and their biological meaning. Different types of milk, not just cow's milk, should be considered. The section on milk fats could be expanded to provide more detailed information on the different types of fatty acids and their effects on health. The importance of trans-access acid (TVA) and its potential health benefits should also be emphasized. What about phospholipid content? In addition to lactose, the topic of milk oligosaccharides could be further developed. The mineral section could add further information on how milk contributes to fulfilling the daily requirements for certain micronutrients.
Positive effects of dairy consumption – It is important to emphasize that the positive effects of milk are not solely due to individual components but are due to the synergistic effects of all their components. The section could be expanded to include a more detailed discussion of the mechanisms of impact bioactive peptides. The section on prebiotics and probiotics could be deeper in terms of the effects of milk oligosaccharides on the intestinal microbiota. It would also be advantageous to use the effects of fermented milk products on the microbiota and the potential health benefits in connection with their consumption. More context is required for CLA. This section could provide a more detailed description of the protective mechanisms against osteoporosis. The cancer section could be more specific in presenting the mechanisms through which milk and dairy products can exert protective effects.
Negative effects of dairy consumption – It is important to note that reactions to dairy products from person to person can vary, depending on factors such as genetics, general health, intestinal bacteria, and nutrition. The type of dairy product is also important. This section could be more detailed on different types of lactose intolerances and their introduction. I suggest a more detailed discussion about the cow's milk protein allergy (CMPA). The section on the possible connection between milk and cancer could be more precise. Consider expanding the topic of the effects on the cardiovascular system. The section on the connection between milk and ASD could be more detailed to explain how beta-casomorphine-7 (beta-cm7), which comes from milk proteins, can influence ASD symptoms. And what about the excess sodium or phosphorus in certain dairy products? What about cholesterol and saturated fat content?
Inconclusive evidence on dairy consumption – It should be noted that reactions to milk consumption can be very individual and depend on many factors, such as genetics, state of health, intestinal microbiota, and general nutrition. In the section, the discrepancies in research results regarding the effects of dairy products on the development of type 2 diabetes could be discussed in more detail. The section on the effects of dairy products on weight could explain the different mechanisms that can influence body mass. The section on the effects of dairy products on the lipid profile and the risk of cardiovascular diseases could be more precise. It should also be noted that most evidence does not support the connection between milk consumption and an increased risk of cardiovascular diseases. The section on the connection between dairy products and prostate cancer risk could be more precise when presenting the contradictory research results. It would be worthwhile to discuss the potential causes of these discrepancies. The section on the association of dairy products with ASD could be expanded to provide more detailed information about the potential mechanism of activity of β-casomorphine-7 (β-CM7) for brain function in people with ASD. It should also be noted that not all people with ASD react negatively to dairy products. I believe there is no tabular summary of the data and evidence discussed.
Milk vs plant-based drinks: the role of milk fat globule membrane – The section could expand the comparison of the nutritional composition of milk and herbal drinks. It is worth noting that there are many types of vegetable drinks and that their nutritional profiles can vary significantly. In my opinion, a tabular summary of the data discussed is missing. It should be emphasized that the effects of nutrients often depend on the food matrix in which they occur, not only on their individual content. Milk is a complex matrix that can contribute to better bioavailability of certain nutrients. The description of MFGM could be more detailed. The section could discuss the documented health benefits of MFGM in more detail. A discussion of the possible disadvantages of MFGM is just as important.
Conclusions – A summary of the advantages of milk and milk product consumption could be more detailed. This section should emphasize more clearly that not all aspects of the effects of milk are clear on health. It would be valuable to add a few words about the effects of the conclusions of the manuscript for nutritional recommendations. The call for further research could be more precise.
Response 1: Many thanks for all your comments. Without any doubt they greatly contribute to improve or manuscript. We have considered your comments and include new information in the sections of the manuscript.